# Telomere Lengths and Serum Proteasome Concentrations in Patients with Type 1 Diabetes and Different Severities of Diabetic Retinopathy in Latvia and Lithuania

**DOI:** 10.3390/jcm11102768

**Published:** 2022-05-13

**Authors:** Zane Svikle, Leonora Pahirko, Līga Zariņa, Kristīne Baumane, Deimante Kardonaite, Lina Radzeviciene, Laura Daugintyte-Petrusiene, Vilma Jurate Balciuniene, Rasa Verkauskiene, Angeļina Tiščuka, Vita Rovite, Nikolajs Sjakste, Jelizaveta Sokolovska

**Affiliations:** 1Faculty of Medicine, University of Latvia, Jelgavas Street 3, LV 1004 Riga, Latvia; zane.svikle@gmail.com (Z.S.); kbaumane75@gmail.com (K.B.); nikolajs.sjakste@lu.lv (N.S.); 2Faculty of Physics, Mathematics and Optometry, University of Latvia, Jelgavas Street 3, LV 1004 Riga, Latvia; leonora.pahirko@lu.lv (L.P.); angellina.ty@gmail.com (A.T.); 3Ophthalmology Department, Riga East University Hospital, Hipokrāta Street 2, LV 1038 Riga, Latvia; bigzars@gmail.com; 4Institute of Endocrinology, Lithuanian University of Health Sciences, 44307 Kaunas, Lithuania; deimante.kardonaite@lsmu.lt (D.K.); lina_radzeviciene@yahoo.com (L.R.); rasa.verkauskiene@lsmuni.lt (R.V.); 5Department of Endocrinology, Medical Academy, Lithuanian University of Health Sciences, 50161 Kaunas, Lithuania; petrusienenator@gmail.com; 6Department of Ophthalmology, Medical Academy, Lithuanian University of Health Sciences, 50161 Kaunas, Lithuania; jurate.balciuniene@kaunoklinikos.lt; 7Latvian Biomedical Research and Study Centre, Rātsupītes Street 1, LV 1067 Riga, Latvia; vita.rovite@biomed.lu.lv

**Keywords:** type 1 diabetes, diabetic retinopathy, diabetic eye disease, telomere, circulating proteasomes

## Abstract

The aim of the study was to compare telomere lengths and circulating proteasome concentrations in patients with different stages of diabetic retinopathy and type 1 diabetes in Latvia and Lithuania. Methods. Patients with no diabetic retinopathy and with non-proliferative diabetic retinopathy were included in the NDR/NPDR group (n = 187). Patients with proliferative diabetic retinopathy and status post laser-photocoagulation were included int the PDR/LPC group (n = 119). Telomeres were evaluated by real-time quantitative polymerase chain reaction. Proteasome concentration was measured by ELISA. Results. Telomeres were longer in PDR/LPC (ΔCT 0.21 (0.12–0.28)) vs. NDR/NPDR (ΔCT 0.18 (0.1–0.28)), *p* = 0.036. In NDR/NPDR, telomeres were correlated negatively with age (R = −0.17, *p* = 0.019), BMI (R = −0.21, *p* = 0.004), waist/hip ratio (R = −0.21, *p* = 0.005), total cholesterol (R = −0.18, *p* = 0.021), and low-density cholesterol (R = −0.20, *p* = 0.010), and positively with estimated glomerular filtration rate (eGFR) (R = 0.28, *p* < 0.001). None of the above correlations were observed in PRD/LPC. Proteasome concentrations were lower in PDR/LPC (130 (90–210) ng/mL) vs. NDR/NPDR (150 (100–240) ng/mL), *p* = 0.024. This correlated negatively with eGFR (R = −0.17, *p* = 0.025) in the NDR/NPDR group and positively with age (R = 0.23, *p* = 0.014) and systolic blood pressure (R = 0.20, *p* = 0.032) in the PRD/LPC group. Telomere lengths did not correlate with proteasome concentrations. Conclusion. Longer telomeres and lower circulating proteasome concentrations are observed in patients with type 1 diabetes and advanced diabetic retinopathy.

## 1. Introduction

Diabetic retinopathy (DR) is a common complication of diabetes mellitus and the leading cause of blindness in the working-age population in the developed world [1]. The pathophysiology of diabetic retinopathy involves dysregulated angiogenesis, increased vascular permeability, vascular occlusion, and neurodegeneration, but the molecular mechanisms behind its pathogenesis are not clear enough [2]. Current therapies for diabetic retinopathy are focused on its advanced stages. Novel biomarkers are needed to improve the screening efficiency and discover novel pharmacotherapeutic targets for the prevention and early treatment of this complication of diabetes [3].

The ubiquitin–proteasome system (UPS) is the main protein quality control system responsible for the recognition and degradation of damaged proteins. The UPS is essential in regulating the cell cycle, immune and inflammatory response, and the degradation of proteins [4]. The UPS is involved in visual functions via the degradation of proteins and the regulation of signalling pathways [5]. Currently available data indicate that hyperglycaemia, increased oxidative stress, and hypoxia disrupt the functions of the UPS [6,7]. These disruptions include the impaired degradation of oxidised proteins [8], endoplasmic reticulum (ER) stress [9], the increased proteasomal degradation of protective and functional proteins (e.g., synaptophysin, rhodopsin, Nrf2, telomerase, telomerase reverse transcriptase) [10,11,12], and the decreased proteasomal degradation of proteins involved in the progression of DR (e.g., HIF1-alpha) [7].

One of the possibilities for studying the UPS in vivo in humans is the measurement of circulating proteasomes—the 20S proteasome—in human blood plasma. In patients suffering from auto-immune rheumatologic disorders, malignancies, sepsis, or trauma, the concentration of circulating proteasomes has been found to be elevated and associated with the disease state and response to treatment [13]. However, there are no data on the levels of circulating proteasomes in type 1 diabetes (T1D) and patients with DR.

Telomeres are tandem repeats (TTAGGG)_n_ of the DNA sequences that form a cap at the ends of eukaryotic chromosomes. Telomeres shorten during each DNA replication cycle until they reach a certain length at which the cell enters senescence and apoptosis. Telomere length is affected by ageing, inflammation, immune system disorders, oxidative stress, genetic as well as environmental factors, and others [14]. An association between shorter telomeres and diabetes [15,16], as well as insulin resistance [17], has been reported in several studies. A recent meta-analysis also demonstrated that the shortening of telomeres is less pronounced in T1D compared to type 2 diabetes (T2D) patients [18]. Several studies have reported an association of telomere length with diabetic kidney disease and macrovascular complications [19,20,21] or the absence of the mentioned association [22]. However, telomere lengths at different stages of diabetic retinopathy have only been reported by one study in patients with T2D [23].

Changes in the proteasomal degradation of proteins with a role in telomere length regulation may affect telomeres. An association between mutations in genes encoding proteasomal subunits, UPS functions and telomere shortening has been observed in several experimental studies [24,25,26] and using System Biology tools [27]. However, an association between telomere length and circulating proteasomes has not been studied in humans with T1D. Taken together, there is a gap in knowledge on telomere length and circulating proteasomes T1D and DR.

In this study, we report the levels of circulating proteasomes and telomere lengths in patients with T1D and different severities of DR from Latvia and Lithuania. Moreover, we describe the cross-sectional association of these biomarkers with anthropometric and clinical measures in T1D.

## 2. Materials and Methods

### 2.1. Patients and Ethics

For this study, in Latvia and Lithuania, patients with T1D were recruited in frames of the international InterDiane consortium. Inclusion criteria for the study were: adult (age ≥ 18 years) patients with a history and established treatment of T1D (defined as an age of diagnosis younger than 40 years, with insulin treatment initiated within one year of diagnosis and C-peptide levels below 0.3 nmol/L) [28]. Exclusion criteria were any diabetes diagnosed after 40 years of age, C-peptide levels above 0.3 nmol/L, and treatment with oral hypoglycaemic agents for more than one year after diagnosis.

The study protocols were approved by the ethics committees in both countries (Latvia: The Latvian Central Ethics Committee; Lithuania: Kaunas Regional Biomedical Research Ethics Committee). In Latvia, the recruitment of the study participants, biobanking, and sample storage were performed in agreement with the procedures of the Genome Database of the Latvian population [29] and are described in more detail in [30]. In Lithuania, patients were recruited from the national T1D registry database and from regular medical visits at the Lithuanian University of Health Sciences Hospital. The study is in line with the ethical standards defined in the 1964 Declaration of Helsinki and its later amendments. Written informed consent was obtained from all study participants prior to inclusion in the study.

### 2.2. Clinical Definitions

DR grading in recruited T1D patients was based on the *fundus oculi* examination performed by an ophthalmologist. Initially, patients were stratified according to the DR status as follows: no retinopathy (NDR), non-proliferative retinopathy (NPDR), proliferative retinopathy (PDR), status after panretinal-laser photocoagulation (LPC). Further, to increase the power of the study, subjects were stratified into two groups.: Patients with no signs of DR and with NPDR were included in the NDR/NPDR group; patients with PDR and status post-laser-photocoagulation were included in the PDR/LPC group.

Albuminuria was assessed using two out of three urine albumin-to-creatinine ratio measurements in the morning spot urine. The estimated glomerular filtration rate (eGFR) was calculated according to Chronic Kidney Disease Epidemiology Collaboration (CKD-EPI).

The definition of diabetic nephropathy was as follows: macroalbuminuria or eGFR below 60 mL/min/1.73 m^2^.

The definition of arterial hypertension was based on blood pressure values and their history of antihypertensive drug usage (patient had hypertension if any of the following applied: systolic blood pressure ≥ 140 mmHg (18.7 kPa) or diastolic blood pressure ≥ 90 mmHg (12.0 kPa), or the regular use of antihypertensive medications).

We defined cardiovascular disease as a history of stroke, amputation, peripheral vascular disease, acute myocardial infarction, or coronary bypass/percutaneous transluminal coronary angioplasty. Smoking was self-reported in the questionnaire, the “smokers” group referred to patients currently smoking at least one cigarette per day.

Body mass index (BMI) was calculated as weight (kg)/height (m)^2^.

### 2.3. Biochemical Parameters

Total cholesterol, low-density lipoprotein and triglycerides, glycated haemoglobin (HbA1c), and the albumin/creatinine ratio in urine were measured in certified clinical laboratories in Latvia and Lithuania.

### 2.4. Sampling of Blood for DNA Extraction and Serum Preparation

For serum preparation, peripheral venous blood was collected. The blood samples were incubated undisturbed for 30 min at room temperature, then centrifuged. The serum was removed from the pellet and transferred into fresh 2 mL tubes, frozen, and stored at −80 °C until analysis. Blood for DNA extraction was collected in EDTA tubes. In Latvia, DNA isolation from whole blood samples using the phenol–chloroform extraction method was carried out in the biobank setting, as previously described [29]. In Lithuania, DNA was extracted by the salting-out method.

### 2.5. Serum Proteasome Measurement

Serum samples from Latvia and Lithuania were assayed in their corresponding countries according to the same protocol. The 20S proteasome concentration was determined by ENZO 20S/26S Proteasome ELISA KIT (Catalogue #: BML-PW0575-0001, Enzo Lifesciences, distributed by AH Diagnostics, Tilst, Denmark). A Wellwash AC plate washer and Multiscan Ascent reader (Thermo Fisher Scientific, Waltham, MA, USA) or “EUROIMMUN Analyzer I” (Version 1.96.0, Euroimmune Medizinische Labordiagnostika AG, Lübeck, Germany) were used. Serum samples were diluted 1:2 in ELISA Buffer before the assay procedure. Standards were diluted as noted in the protocol: 1.6 μg/mL, 0.8 μg/mL, 0.4 μg/mL, 0.2 μg/mL, 0.1 μg/mL, 0.05 μg/mL, 0.025 μg/mL, and 0 μg/mL (S0). The calculation of the proteasome concentrations in the samples was handled by an immunoassay software package (ENZO Assay Blaster, Enzo Lifesciences, distributed by AH Diagnostics, Tilst, Denmark) utilising a 4-parameter logistic curve fitting program.

### 2.6. Telomere Length Detection

Samples from Latvia and Lithuania were assayed in their corresponding countries according to the same protocol. The analysis was performed on a StepOnePlus™ real-time PCR system (Applied Biosystems, Thermo Fisher Scientific, Waltham, MA, USA) or a “Rotor-Gene Q” (QIAGEN, Hilden, Germany) real-time PCR cycler. The relative telomere length method (DCT) described by Zole et al. [31] was performed to measure telomere length in DNA samples from T1D patients, using beta-globin as a reference gene. qPCR was performed using Maxima SYBR Green qPCR Master Mix (2×) (Thermo Fisher Scientific, Waltham, MA, USA). The forward and reverse primers for performing telomere qPCR in one reaction were: Telo1 (200 nM), 5′-GGTTTTTGAGGGTGAGGGTGAGGGTGAGGGTGAG-GGT-3′;

Telo2 (200 nM), 5′-CCCGACTATCCCTATCCCTATCCCTATCCCTATCCCTA-3′.

A qRT-PCR for telomere length detection was performed under the following conditions: a denaturation step at 95 °C for 10 min; the DNA samples were then incubated for 40 cycles at 10 s at 95 °C and 1 min at 58 °C. Efficiency = 92%. For telomere length normalisation, the following forward and reverse primers for the b-globin gene in a separate run were used:

Beta-glob1 (300 nM), 5′-GCTTCTGACACAACTGTGTTCACTAGC-3′;

Beta-glob2 (500 nM), 5′-CACCAACTT CATCCACGTTCACC-3′.

Reaction conditions for beta-globin were: a denaturation step at 95 °C for 10 min, incubation for 40 cycles at 95 °C for 10 s, and at 56 °C for 20 s [32]. Efficiency = 90%. The concentration of the DNA was 5 ng/μL in a 15 μL reaction. Each sample was run in triplicate. A no-template control and duplicate calibrator samples were used in all runs to allow for a comparison of the results across all runs. A melting curve analysis was performed to verify the specificity and identity of the PCR products. Telomere length was calculated using threshold cycle or CT values and the equation: telomere length ratio (test/reference) = 2^Ct(bglobin)−Ct(telomeres)^.

### 2.7. Statistical Analysis

Most of the variables analysed did not correspond with the normality assumption (according to the Shapiro–Wilk test); therefore, data are presented as medians with q0.25–q0.75. Data analysis was carried out on a country-by-country basis for Latvia and Lithuania, as well as for the whole cohort. The Wilcoxon rank-sum test was initially used to compare the distributions of biomarkers between the groups of DR (NDR/NPDR and PDR/LPC). Diabetes duration, waist/hip ratio, serum triglycerides, and eGFR differed statistically significantly between the NDR/NPDR and PDR/LPC groups. Therefore, a one-way analysis of covariance (ANCOVA) on ranks was performed to determine statistically significant differences in biomarkers controlling for these covariates. Gender distribution was not significantly different between NDR/NPDR and PDR/LPC groups, therefore it was not used as the covariate.

Correlation analysis was carried out on a country-by-country basis using the Spearman correlation coefficient. For the whole cohort analysis, standardised values (Z-scores) of biomarkers were calculated and then pooled.

Logistic regression models were fitted to evaluate the odds of severe diabetic retinopathy associated with serum proteasome concentration and telomere length. Model 1 was adjusted for age, sex, and BMI; Model 2 was additionally adjusted for diabetes duration, waist/hip ratio, HbA1c, arterial hypertension, total cholesterol, high-density lipoprotein cholesterol, and triglycerides. Covariates were selected according to the statistically significant differences in biomarkers between DR groups. Models were constructed separately for Latvian and Lithuanian cohorts, as well as for the whole cohort. In the latter case, all models were additionally adjusted for the country.

Statistical data analysis was performed using Statistical Software R (R Development Core Team).

## 3. Results

### 3.1. Characteristics of Latvian and Lithuanian Cohorts

186 patients from Latvia and 120 patients from Lithuania were included in the study. Age, BMI, waist/hip ratio, systolic blood pressure, and diastolic blood pressure did not differ statistically significantly in both cohorts. On the contrary, in comparison to the Latvian cohort, Lithuanian patients had statistically significantly shorter diabetes duration, a lower prevalence of cardiovascular disease, arterial hypertension, statin usage, lower eGFR levels, and higher total cholesterol and low-density lipoprotein. In the Latvian cohort, there were 69 patients with NDR, 38 patients with NPDR and 79 patients with PDR/LPC. In Lithuania, the above groups were represented by 39, 41, and 40 patients, respectively (Table 1, Appendix A).

### 3.2. Characteristics of Patients with Different Severity of DR

As seen in Table 2, patients with PDR/LPC were significantly older, with a longer diabetes duration, arterial hypertension and other complications of diabetes, higher serum triglycerides concentration, and lower eGFR.

### 3.3. Telomere Length in Patients with Different Severity of DR

The median telomere length in the Latvian cohort was statistically significantly shorter compared to the median in the Lithuanian cohort (Latvia: 0.14 (0.09–0.26); Lithuania: 0.24 (0.18–0.28); *p* < 0.001). In a country-by-country analysis, both in Latvia and Lithuania, telomere length did not differ statistically significantly in patients with NDR/NPDR and PDR/LPC (Latvia: NDR/NPDR 0.13 (0.07–0.26), PDR/LPC 0.17 (0.1–0.26), *p* = 0.135; Lithuania NDR/NPDR 0.22 (0.16–0.28), PDR/LPC 0.26 (0.19–0.28), *p* = 0.281). Adjustment for confounders did not change the results significantly. However, when the respective groups of both countries were pooled, and after adjusting for the country, duration of diabetes, waist/hip ratio, serum triglycerides, and eGFR, we observed that telomeres were statistically significantly longer in patients with PDR/LPC (ΔCT: PDR/LPC 0.21 (0.12–0.28) vs. NDR/NPDR 0.18 (0.1–0.28); adjusted mean ranks: PDR/LPC 160 (113.9–206), NDR/NPDR 134 (94.3–173), *p* = 0.036). See Table 2 and Figure 1.

### 3.4. Serum Proteasome Concentration in Patients with Different Severity of DR

The median proteasome concentration in the Latvian cohort was statistically significantly lower compared to the Lithuanian cohort (Latvia: 120 (80–200) ng/mL; Lithuania: 150 (120–240) ng/mL; *p* < 0.001). In a country-by-country analysis, both in Latvia and Lithuania, serum proteasome concentrations did not differ statistically significantly between patients with NDR/NPDR and PDR/LPC (Latvia: NDR/NPDR 120 (80–200) ng/mL, PDR/LPC 120 (77.5–192.5) ng/mL, *p* = 0.423; Lithuania NDR/NPDR 165 (120–240) ng/mL, PDR/LPC 150 (120–213.75) ng/mL, *p* = 0.401). Adjustment for confounders did not change the results significantly. However, when the respective groups from both countries were pooled, and after adjusting for the country, duration of diabetes, waist/hip ratio, serum triglycerides, and eGFR, we observed that serum proteasome concentrations were statistically significantly lower in patients with PDR/LPC (PDR/LPC 130 (90–210) ng/mL vs. NDR/NPDR 150 (100–240) ng/mL; adjusted mean ranks: PDR/LPC 106 (58.8–152), NDR/NPDR 134 (94.2–174), *p* = 0.023). See Table 2 and Figure 1.

### 3.5. Correlations between Telomere Length, Proteasome Concentration, and Clinical Parameters

In the whole cohort, a negative statistically significant correlation was observed between telomere length and lipids (total cholesterol: R = −0.15, *p* = 0.010, low-density lipoprotein: R = −0.17, *p* = 0.004). However, when patients were stratified according to the severity of diabetic retinopathy, distinct biomarker correlation patterns were observed in NDR/NPDR and PRD/LPC. In the NDR/NPDR group, negative correlations were observed between telomere length and age (R = −0.17, *p* = 0.019), BMI (R = −0.21, *p* = 0.004), waist/hip ratio (R = −0.21, *p* = 0.005), total cholesterol (R = −0.18, *p* = 0.021), and low-density lipoproteins (R = −0.20, *p* = 0.010), and a positive correlation was observed with eGFR (R = 0.28, *p* < 0.001). On the contrary, no significant correlations between telomere length and the above clinical parameters were found in the PRD/LPC group in the whole-cohort analysis. Similar differences in biomarker correlation patterns with telomere length between NDR/NPDR and PRD/LPC groups were also observed in the separate country analyses (see Table 3).

In the whole cohort, a positive statistically significant correlation was observed between serum proteasome concentration and systolic blood pressure (R = 0.16, *p* = 0.008). In the NDR/NPDR group, serum proteasome concentration was negatively correlated with eGFR (R = −0.17, *p* = 0.025). In the PRD/LPC group, serum proteasome concentration was positively correlated with age (R = 0.23, *p* = 0.014) and systolic blood pressure (R = 0.20, *p* = 0.032). Similar differences in biomarker correlation patterns with proteasome concentrations between the NDR/NPDR and PRD/LPC groups were also observed in separate country analyses (see Table 4).

### 3.6. Association of Severe Diabetic Retinopathy with Telomere Length and Proteasome Concentration

In logistic regression models, longer telomeres and lower serum proteasome concentrations increased the odds of PRD/LPC; however, they were not statistically significant in the majority of tested models. In whole cohort analysis, the results of Model 2 demonstrate a statistically significant association of serum proteasome concentration with lower odds of PRD/LPC when both telomere length and proteasome concentration were used as predictors (odds ratio: 0.35, 95% CI: (0.13, 0.97), *p* = 0.042) (Table 5).

## 4. Discussion

In this study, we report longer telomeres and lower serum proteasome concentrations in a cohort of 306 Latvian and Lithuanian patients with T1D and advanced stages of diabetic retinopathy (PRD/LPC). Moreover, we report differences in the correlation patterns of anthropometric and clinical biomarkers with telomere lengths and proteasome concentrations between NDR/NPDR and PRD/LPC groups.

We observed shorter telomeres and lower serum proteasome concentrations in Latvian patients compared to Lithuanian patients. The Latvian and Lithuanian cohorts differed in several factors previously associated with telomere length, such as the duration of diabetes, the prevalence of arterial hypertension and cardiovascular disease, etc. [33]. In addition, although Latvia and Lithuania are neighbouring countries and both refer to the Baltic region, genetic differences in both populations have been observed previously [34]. The environment is another factor that could affect both telomere length and proteasome concentration via telomerase activity [35] and UPS activity [36]. Therefore, adjustment for the country was used in all comparisons of telomere lengths and proteasome concentrations between NDR/NPDR and PRD/LPC groups, and pooled Z-scores were used for correlation analysis in these groups.

In contrast to the abundance of studies reporting telomere lengths in people with T1D and T2D compared to generally healthy adults, only a few studies have addressed telomeres in relation to the microvascular complications of diabetes. To our knowledge, only one study reports telomere length in patients with different retinopathy stages and T2D. In that study by Sharma et al., telomeres were found to be shorter in subjects with more severe forms of DR compared to patients without DR [23]. In contrast to the above results, we found that telomeres are longer in patients with T1D and PRD/LPC compared to NDR/NPDR. The difference could be explained by the fact that T2D and T1D are two different diseases. In addition, subjects included in the study by Sharma et al., had very good diabetes control (HbA1c < 7%) and were older (age 45.5–60.5 years), while the duration of diabetes was not reported. In contrast, our study includes relatively young adults with long-standing T1D and bad glycaemic control.

Previously, age, diabetes duration and waist/hip ratio were shown to be negatively associated with telomere length in T2D [33]. As Latvian and Lithuanian patients with PRD/LPC had a longer diabetes duration, older age, and higher waist/hip ratio compared to the NDR/NPDR group, one would expect to observe shorter telomeres in the PRD/LPC group, but this was not the case in our study. Our analysis was adjusted for these factors, therefore, we report the actual association between severe retinopathy and telomere lengths in T1D, indicating no evidence of telomere shortening in advanced retinopathy.

Moreover, the correlation analysis performed in our study supports the above results on the absence of associations between “traditional” factors for telomere shortening and their actual length in advanced diabetic retinopathy in T1D. Indeed, in the NDR/NPDR group, negative correlations were observed between telomere length and age, BMI, waist/hip ratio, total cholesterol, low-density lipoproteins, and a positive correlation with eGFR. The observed pattern is very much in agreement with previously published data on the general population and T2D, as cardiovascular risk factors such as obesity and insulin resistance were associated with shorter telomeres in several studies [37,38,39]. The absence of an association between the above parameters and telomere lengths in the PRD/LPC group might indicate the loss of “traditional” patterns in telomere length regulation in advanced microvascular disease in T1D.

In our cohort, among possible factors that might influence telomere lengthening in patients with advanced retinopathy and T1D, a higher prevalence of statins and antihypertensive medication in PRD/LPC usage are the first on the list. In the general population, statins have been shown to increase telomerase activity [40] and antihypertensive medication usage was demonstrated to be protective for telomeres [33].

A possible factor that could be associated with longer telomeres in the PRD/LPC group is lifestyle. A study in patients with prediabetes demonstrated telomere lengthening over a period of 4.5 years due to lifestyle intervention [39]. Patients with complications of diabetes might be more motivated to adopt a healthier lifestyle to avoid the progression of complications. However, apart from smoking (which was not associated with telomere length in our study), lifestyle was out of the scope of our study.

Another factor for telomere elongation could be the telomere-independent functions of telomerase. For example, telomerase directly regulates NF-κB-dependent gene expression, such as that of IL6 and TNFα [41]; cytokines were previously shown to be involved in the inflammatory pathogenetic pathways of the complications of diabetes [2,42] and cancer [43]. Both diabetes and cancer are characterised by chronic hypoxia [11,44]. Mild prolonged hypoxia was shown to increase telomerase activity and elongate telomeres in vitro [45] and in humans [46].

Our findings on the serum levels of circulating 20S proteasomes demonstrate that serum proteasome concentration is lower in patients with T1D and PRD/LPC compared to NDR/NPDR in the whole cohort adjusted for country. The level of circulating proteasomes has not been previously reported in T1D. However, higher circulating proteasome levels were reported in patients with auto-immune rheumatologic disorders, malignancies, and inflammatory states; an increase in serum proteasome levels in the mentioned disorders is considered a protective reaction in response to disease flare-up [13]. In our study, lower circulating proteasome concentrations in patients with advanced DR might be associated with the pathogenesis of chronic T1D microvascular complications. Indeed, DR is a slowly progressive disorder, often silent and undiagnosed until severe sight disturbances occur. It is associated with chronic hyperglycaemia, oxidative stress, hypoxia, and low-grade inflammation [47]. Previous studies have indicated that hyperglycaemia may decrease proteasomal activity [48]. The state of oxidative stress, which is typical for diabetes, was also shown to interfere with proteasomal activity and increase the amount of ubiquitinated protein in cell cultures [49]. In addition, proteasome activity is impaired in response to hyperglycaemia-associated hypoxia [6]. Derangements in proteasome activity related to hyperglycaemia, oxidative stress, and hypoxia may contribute to the progression of diabetic retinopathy via changes in the degradation of oxidised proteins, ER stress, and the decreased proteasomal degradation of proteins involved in the progression of DR (e.g., HIF1-alpha) [6,7,8,9].

In our study, telomere length and serum proteasome concentration were not significantly associated. However, our finding of lower serum proteasome concentration and higher telomere length in patients with PRD/LPC may indirectly support the previous findings on the inverse association between UPS and telomerase activity [16].

The limitations of the study include its cross-sectional nature and relatively low number of patients included, especially in the separate country analyses. In addition, the analyses of telomere lengths and serum proteasome concentrations, although performed by the same methods, were done in two different centres. We tried to alleviate the latter limitation by the adjustment of the whole cohort analysis for the country and the utilisation of Z-scores for the correlation analysis. The main strength of the study lies in the fact that it is the first study reporting association between telomere length, serum proteasome concentration, and stages of DR in T1D.

To conclude, longer telomeres and lower serum proteasome concentrations are observed in patients with T1D and advanced DR as compared to patients without the presence of DR or in its initial stage.

## Figures and Tables

**Figure 1 jcm-11-02768-f001:**
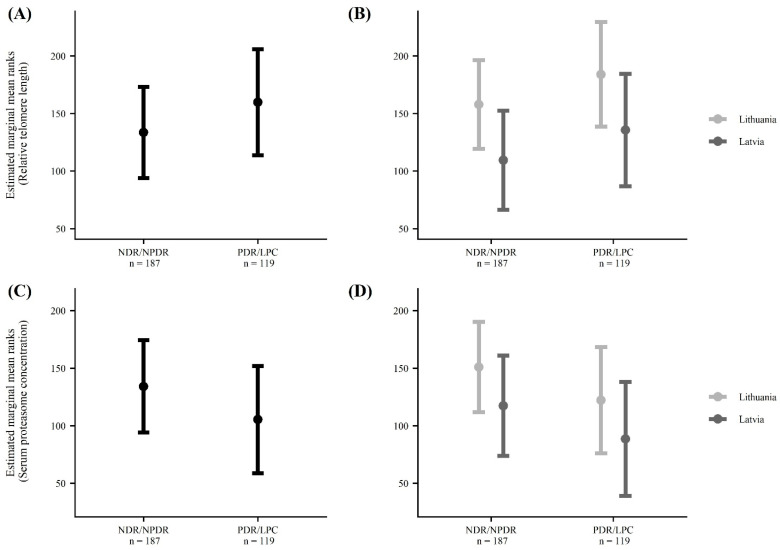
Telomere length and serum proteasome concentration in patients with different severities of diabetic retinopathy in Latvia and Lithuania. NDR/NPDR—no diabetic retinopathy/non-proliferative retinopathy; PDR/LPC—proliferative retinopathy/status after panretinal-laser photocoagulation. (**A**) Telomere length in the whole cohort was stratified according to diabetic retinopathy status. *p* = 0.036 between NDR/NPDR and PDR/LPC after controlling for the country, duration of diabetes, waist/hip ratio, serum triglycerides and eGFR. (**B**) Telomere length in patients with different severities of diabetic retinopathy stratified by country. (**C**) Circulating proteasome concentrations in the whole cohort stratified according to diabetic retinopathy status. *p* = 0.0234 between NDR/NPDR and PDR/LPC after controlling for the country, duration of diabetes, waist/hip ratio, serum triglycerides and eGFR. (**D**) Circulating proteasome concentration in patients with different severities of diabetic retinopathy stratified by country.

**Table 1 jcm-11-02768-t001:** Characteristics of the subjects.

	Whole Cohort, n = 306	Latvia, n = 186	Lithuania, n = 120	*p*-Value *
Age, years	37 (27–48)	38 (28–49)	35 (25–47)	0.11
Male/female, n (%)	162/144 (53/47)	90/96 (48/52)	72/48 (60/40)	0.06
BMI, kg/m^2^	24.8 (22.2–28.1)	25.0 (22.5–28.5)	24.1 (22.0–27.7)	0.12
Waist/hip ratio	0.85 (0.78–0.92)	0.85 (0.78–0.91)	0.84 (0.78–0.93)	0.95
Systolic blood pressure, mmHg	126.5 (117.9–138.0)	126.0 (118.0–138.0)	127.5 (117.5–137.5)	0.84
Diastolic blood pressure, mmHg	80.0 (74.0–89.0)	82.5 (74.2–89.8)	80.0 (74.2–87.8)	0.25
Arterial hypertension, n (%)	174 (57)	115 (62)	59 (49)	0.04
Duration of diabetes, years	20 (14–28)	21 (15–31)	19 (13–27)	0.03
Diabetic nephropathy, n (%)	43 (14)	25 (13)	18 (15)	0.47
NDR, n (%)	108 (35)	69 (37)	39 (33)	0.14
NPDR, n (%)	79 (26)	38 (20)	41 (34)	
PDR/LPC, n (%)	119 (39)	79 (42)	40 (33)	
Cardiovascular disease, n (%)	31 (10)	26 (14)	5 (4)	0.001
Polyneuropathy, n (%)	201 (66)	112 (60)	89 (74)	0.02
Smoking, n (%)	69 (23)	49 (26)	20 (17)	0.07
HbA1c, %	8.5 (7.5–9.8)	8.5 (7.7–9.9)	8.4 (7.3–9.6)	0.17
HbA1c, mmol/mol	69.4 (58.7–83.6)	69.4 (60.7–84.7)	68.3 (56.8–81.4)	0.17
Total cholesterol, mmol/L	4.9 (4.2–5.8)	4.8 (4.0–5.7)	5.2 (4.5–5.8)	0.01
High-density lipoprotein, mmol/L	1.6 (1.3–1.9)	1.5 (1.2–1.8)	1.6 (1.3–2.0)	0.13
Low-density lipoprotein, mmol/L	2.8 (2.2–3.4)	2.6 (2.0–3.3)	2.9 (2.5–3.5)	0.005
Triglycerides, mmol/L	1.0 (0.7–1.4)	1.1 (0.7–1.7)	0.9 (0.7–1.2)	0.001
eGFR, mL/min/1.73 m^2^	110 (87–122)	112 (93–125)	100 (80–120)	0.009
Statins usage, n (%)	50 (16)	41 (22)	9 (8)	0.001
ACEI/ARB usage, n (%)	101 (33)	65 (35)	36 (30)	0.44
Serum proteasome concentration, ng/mL	140 (90–218)	120 (80–200)	150 (120–240)	<0.001
Relative telomere length	0.19 (0.11–0.28)	0.14 (0.09–0.26)	0.24 (0.18–0.28)	<0.001

Continuous data are presented as median with q0.25–q0.75 and categorical data are presented as frequencies. * *p*-value of Wilcoxon rank-sum test for continuous variables or Chi-squared test for the equality of proportions for categorical variables between Latvia and Lithuania. Nephropathy, if macroalbuminuria, ESRD or eGFR < 60 mL/min/1.73 m^2^. BMI—body mass index; NDR/NPDR—no diabetic retinopathy/non-proliferative retinopathy; PDR/LPC—proliferative retinopathy/status after panretinal-laser photocoagulation; eGFR—estimated glomerular filtration rate (CKD-EPI); ACEI/ARB—angiotensin-converting enzyme inhibitors/angiotensin II receptor blockers.

**Table 2 jcm-11-02768-t002:** Characteristics of patients stratified according to retinopathy status in Latvia, Lithuania, and the whole cohort.

	Latvia			Lithuania			Whole Cohort		
	NDR/NPDR n = 107	PDR/LPC n = 79	*p*-Value	NDR/NPDR n = 80	PDR/LPC n = 40	*p*-Value	NDR/NPDR n = 187	PDR/LPC n = 119	*p*-Value
Age, years	32 (26–43)	46 (38–55)	<0.001	31 (24–47)	41 (33–48)	0.004	31 (25–44)	44 (35–54)	<0.001
Male/female, n (%)	55/52 (51/49)	35/44 (44/56)	0.419	47/33 (59/41)	25/15 (63/37)	0.843	102/85 (55/45)	60/59 (50/50)	0.557
Body mass index, kg/m^2^	24.8 (22.2–28.0)	25.6 (23.4–29.0)	0.198	24.1 (22.0–27.7)	24.5 (21.9–28.1)	0.654	24.5 (22.2–27.9)	25.5 (22.5–28.6)	0.143
Waist/hip ratio	0.84 (0.78–0.90)	0.86 (0.79–0.94)	0.120	0.82 (0.77–0.92)	0.88 (0.80–0.93)	0.070	0.83 (0.77–0.90)	0.87 (0.79–0.94)	0.019
Systolic blood pressure, mmHg	126 (118–135)	126 (119–146)	0.347	126 (118–138)	128 (120–142)	0.351	126 (118–138)	127 (119–143)	0.217
Diastolic blood pressure, mmHg	82 (75–89)	83 (74–90)	0.988	80 (75–88)	80 (74–87)	0.758	80 (75–88)	80 (74–90)	0.837
Arterial hypertension, n (%)	49 (46)	66 (84)	<0.001	30 (38)	29 (73)	<0.001	79 (42)	95 (80)	<0.001
Duration of diabetes, years	17 (13–22)	31 (22–36)	<0.001	15 (11–21)	27 (21–33)	<0.001	16 (12–22)	28 (22–36)	<0.001
Diabetic nephropathy, n (%)	7 (7)	18 (23)	0.003	8 (10)	10 (25)	0.058	15 (8)	28 (24)	<0.001
Cardiovascular disease, n (%)	2 (2)	24 (30)	<0.001	4 (5)	1 (3)	0.872	6 (3)	25 (21)	<0.001
Polyneuropathy, n (%)	54 (50)	58 (73)	0.003	51 (64)	38 (95)	<0.001	105 (56)	96 (81)	<0.001
Smoking, n (%)	36 (34)	13 (16)	0.014	9 (11)	11 (28)	0.046	45 (24)	24 (20)	0.513
HbA1c, %	8.6 (7.9–10.2)	8.4 (7.6–9.6)	0.143	8.4 (7.3–9.7)	8.5 (7.7–9.7)	0.529	8.5 (7.5–9.9)	8.4 (7.6–9.6)	0.508
HbA1c, mmol/mol	70.0 (62.3–88.0)	67.8 (59.6–81.4)	0.143	68.3 (56.3–81.0)	69.4 (60.3–82.0)	0.529	69.4 (58.5–84.7)	67.8 (59.8–81.4)	0.508
Total cholesterol, mmol/L	4.6 (4.0–5.7)	4.9 (3.8–5.8)	0.724	5.0 (4.4–5.8)	5.4 (4.8–6.4)	0.027	4.8 (4.1–5.7)	5.2 (4.2–5.8)	0.186
HDL, mmol/L	1.5 (1.2–1.8)	1.5 (1.2–2.0)	0.469	1.6 (1.3–2.0)	1.5 (1.3–1.9)	0.709	1.6 (1.2–1.8)	1.5 (1.3–1.9)	0.506
LDL, mmol/L	2.6 (2.1–3.3)	2.6 (2.0–3.3)	0.754	2.8 (2.3–3.3)	3.3 (2.7–3.9)	0.007	2.7 (2.2–3.3)	2.9 (2.2–3.5)	0.325
Triglycerides, mmol/L	1.0 (0.7–1.5)	1.2 (0.8–1.7)	0.067	0.8 (0.6–1.1)	1.1 (0.7–1.3)	0.045	0.9 (0.7–1.3)	1.1 (0.8–1.6)	0.003
eGFR, mL/min/1.73 m^2^	119 (110–128)	99 (76–113)	<0.001	106 (86–123)	93 (71–111)	0.007	115 (99–126)	98 (72–112)	<0.001
Statin usage, n (%)	7 (7)	34 (43)	<0.001	4 (5)	5 (12.5)	0.270	11 (6)	39 (33)	<0.001
ACEI/ARB usage, n (%)	22 (21)	43 (54)	<0.001	15 (19)	21 (52.5)	<0.001	37 (20)	64 (54)	<0.001
Serum proteasome concentration, ng/mL	120 (80–200)	120 (77.5–192.5)	0.423	165 (120–240)	150 (120–213.75)	0.401	150 (100–240)	130 (90–210)	0.023 *
Relative telomere length	0.13 (0.07–0.26)	0.17 (0.1–0.26)	0.135	0.22 (0.16–0.28)	0.26 (0.19–0.28)	0.281	0.18 (0.1–0.28)	0.21 (0.12–0.28)	0.036 *

Continuous data are presented as medians with q0.25–q0.75 with the corresponding Wilcoxon rank-sum test *p*-value; *—ANCOVA on ranks adjusted for diabetes duration, waist/hip ratio, serum triglycerides, eGFR, and additionally adjusted for the participating country in the whole cohort analysis was performed. Categorical data are presented as frequencies with the corresponding *p*-values from the Chi-squared test for equality of proportions. NDR/NPDR—no diabetic retinopathy/non-proliferative retinopathy; PDR/LPC—proliferative retinopathy/status after panretinal-laser photocoagulation; eGFR—estimated glomerular filtration rate (CKD-EPI); ACEI/ARB—angiotensin-converting enzyme inhibitors/angiotensin II receptor blockers.

**Table 3 jcm-11-02768-t003:** Correlations between relative telomere length and clinical markers.

	All Patients, R (*p*-Value)			NDR/NPDR, R (*p*-Value)			PDR/LPC, R (*p*-Value)		
	Latvia, n = 186	Lithuania, n = 120	Whole Cohort, n = 306	Latvia, n = 107	Lithuania, n = 80	Whole Cohort, n = 187	Latvia, n = 79	Lithuania, n = 40	Whole Cohort, n = 119
**Age**	–0.02 (0.82)	–0.15 (0.104)	–0.04 (0.449)	**−0.22 (0.022)**	–0.21 (0.056)	**−0.17 (0.019)**	0.18 (0.113)	–0.18 (0.264)	0.08 (0.366)
**Body mass index**	–0.02 (0.799)	**−0.19 (0.041)**	–0.6 (0.268)	**−0.24 (0.015)**	**−0.23 (0.045)**	**−0.21 (0.004)**	**0.25 (0.028)**	–0.15 (0.355)	0.14 (0.138)
**Waist/hip ratio**	–0.05 (0.489)	–0.13 (0.153)	–0.7 (0.212)	**−0.21 (0.028)**	**−0.22 (0.050)**	**−0.21 (0.005)**	0.14 (0.235)	0.01 (0.973)	0.11 (0.260)
**Systolic blood pressure**	**−0.16 (0.026)**	0.04 (0.704)	–0.8 (0.195)	–0.18 (0.063)	0.1 (0.387)	–0.05 (0.543)	–0.14 (0.219)	–0.11 (0.491)	–0.13 (0.177)
**Diastolic blood pressure**	–0.12 (0.113)	–0.15 (0.106)	–0.11 (0.053)	–0.09 (0.378)	–0.16 (0.155)	–0.10 (0.180)	–0.17 (0.148)	–0.12 (0.451)	–0.14 (0.126)
**Duration of diabetes**	0.07 (0.328)	–0.08 (0.397)	0.02 (0.676)	0.01 (0.932)	**−0.27 (0.015)**	–0.11 (0.129)	0.07 (0.523)	–0.04 (0.796)	0.07 (0.432)
**HbA1c**	–0.03 (0.673)	**0.26 (0.005)**	0.08 (0.193)	–0.05 (0.616)	**0.35 (0.002)**	0.11 (0.152)	0.1 (0.394)	0.08 (0.644)	0.08 (0.413)
**Total cholesterol**	**−0.18 (0.023)**	–0.14 (0.135)	**−0.15 (0.010)**	**−0.25 (0.014)**	–0.11 (0.345)	**−0.18 (0.021)**	–0.08 (0.512)	–0.23 (0.147)	–0.13 (0.160)
**High-density lipoproteins**	–0.1 (0.213)	–0.01 (0.873)	–0.05 (0.420)	–0.12 (0.259)	0.06 (0.612)	–0.01 (0.900)	–0.09 (0.452)	–0.14 (0.388)	–0.14 (0.254)
**Low-density lipoproteins**	**−0.19 (0.013)**	–0.16 (0.088)	**−0.17 (0.004)**	**−0.21 (0.037)**	–0.18 (0.12)	**−0.20 (0.010)**	–0.14 (0.237)	–0.22 (0.177)	–0.14 (0.145)
**Triglycerides**	–0.11 (0.142)	–0.09 (0.361)	–0.09 (0.135)	–0.2 (0.059)	–0.03 (0.784)	–0.11 (0.156)	–0.04 (0.717)	**−0.32 (0.047)**	–0.10 (0.294)
**eGFR**	0.01 (0.945)	**0.21 (0.026)**	0.09 (0.121)	**0.2 (0.047)**	**0.4 (<0.001)**	**0.28 (<0.001)**	–0.05 (0.638)	–0.12 (0.478)	–0.05 (0.570)
**Serum proteasome concentration**	–0.06 (0.45)	0.03 (0.711)	–0.03 (0.667)	–0.06 (0.539)	0.07 (0.556)	–0.02 (0.833)	–0.01 (0.924)	–0.03 (0.865)	–0.02 (0.796)

Data are presented as Spearman correlation coefficient R (*p*-value). Correlations were calculated on raw biomarkers for the corresponding country analysis. In the whole cohort analysis, pooled Z-scores were used for correlation analysis. Entries with *p* < 0.05 are highlighted in bold. NDR/NPDR—no diabetic retinopathy/non-proliferative retinopathy; PDR/LPC—proliferative retinopathy/status after panretinal-laser photocoagulation; eGFR—estimated glomerular filtration rate (CKD-EPI).

**Table 4 jcm-11-02768-t004:** Correlations between serum proteasome concentrations and clinical markers.

	All Patients, R (*p*-Value)			NDR/NPDR, R (*p*-Value)			PDR/LPC Group, R (*p*-Value)		
	Latvia, n = 186	Lithuania, n = 120	Whole Cohort, n = 306	Latvia, n = 107	Lithuania, n = 80	Whole Cohort, n = 187	Latvia, n = 79	Lithuania, n = 40	Whole Cohort, n = 119
**Age**	−0.02 (0.754)	**0.25 (0.006)**	0.11 (0.055)	−0.02 (0.809)	0.16 (0.150)	0.09 (0.232)	−0.01 (0.917)	**0.58 (<0.001)**	**0.23 (0.014)**
**Body mass index**	0.08 (0.277)	0.10 (0.306)	0.09 (0.110)	−0.05 (0.609)	0.14 (0.215)	0.06 (0.413)	**0.27 (0.019)**	0.02 (0.889)	0.17 (0.076)
**Waist/hip ratio**	0.13 (0.088)	0.08 (0.391)	0.11 (0.069)	0.09 (0.369)	0.06 (0.593)	0.08 (0.297)	0.21 (0.077)	0.17 (0.296)	0.18 (0.055)
**Systolic blood pressure**	0.01 (0.930)	**0.33 (<0.001)**	**0.16 (0.008)**	−0.04 (0.687)	**0.32 (0.004)**	0.14 (0.061)	0.07 (0.569)	**0.4 (0.011)**	**0.20 (0.032)**
**Diastolic blood pressure**	−0.01 (0.917)	−0.04 (0.637)	−0.02 (0.738)	−0.11 (0.262)	−0.01 (0.902)	−0.05 (0.545)	0.12 (0.294)	−0.07 (0.656)	0.02 (0.800)
**Duration of diabetes**	0.03 (0.706)	0.05 (0.559)	0.04 (0.490)	0.14 (0.168)	0.00 (0.995)	0.07 (0.377)	−0.07 (0.558)	**0.34 (0.029)**	0.11 (0.237)
**HbA1c**	−0.04 (0.609)	−0.01 (0.947)	−0.03 (0.612)	−0.06 (0.536)	0.03 (0.803)	−0.03 (0.692)	−0.01 (0.932)	−0.08 (0.633)	−0.05 (0.639)
**Total cholesterol**	−0.04 (0.632)	−0.04 (0.709)	−0.05 (0.450)	−0.06 (0.549)	0.06 (0.595)	−0.03 (0.729)	0.01 (0.937)	−0.17 (0.295)	−0.06 (0.556)
**High-density lipoproteins**	−0.10 (0.195)	−0.04 (0.693)	−0.07 (0.261)	−0.11 (0.316)	−0.03 (0.787)	−0.07 (0.401)	−0.12 (0.314)	−0.05 (0.747)	−0.08 (0.379)
**Low-density lipoproteins**	0.01 (0.898)	−0.07 (0.438)	−0.04 (0.497)	0.01 (0.906)	0.07 (0.526)	0.01 (0.912)	0.00 (0.985)	−0.28 (0.076)	−0.12 (0.190)
**Triglycerides**	−0.01 (0.924)	0.07 (0.456)	0.03 (0.609)	−0.07 (0.491)	0.02 (0.894)	−0.03 (0.687)	0.08 (0.485)	0.22 (0.178)	0.14 (0.130)
**eGFR**	−0.03 (0.704)	**−0.20 (0.030)**	−0.11 (0.053)	−0.10 (0.307)	−0.21 (0.062)	**−0.17 (0.025)**	0.02 (0.889)	**−0.33 (0.045)**	−0.12 (0.191)
**Relative telomere length**	−0.06 (0.450)	0.03 (0.711)	−0.03 (0.667)	−0.06 (0.539)	0.07 (0.556)	−0.02 (0.833)	−0.01 (0.924)	−0.03 (0.865)	−0.02 (0.796)

Data are presented as Spearman correlation coefficient R (*p*-value). Correlations were calculated on raw biomarkers for the country analysis. In the whole cohort analysis, pooled Z-scores were used for correlation analysis. Entries with *p* < 0.05 are highlighted in bold. NDR/NPDR—no diabetic retinopathy/non-proliferative retinopathy; PDR/LPC—proliferative retinopathy/status after panretinal-laser photocoagulation. eGFR—estimated glomerular filtration rate (CKD-EPI).

**Table 5 jcm-11-02768-t005:** Association between severe retinopathy, telomeres and proteasomes on a country-by-country basis and in the whole cohort.

Predictor		Relative Telomere Length Ratio		ELISA Proteasome Concentration		Relative Telomere Length Ratio + ELISA Proteasome Concentration	
		Odds Ratio (95% CI)	*p*-Value	Odds Ratio (95% CI)	*p*-Value	Odds Ratio (95% CI)	*p*-Value
**Latvia**	**Model 1**	2.04 (0.44, 9.45)	0.361	0.66 (0.38, 1.14)	0.132	***TL*** 2.79 (0.53, 14.66) ***PR*** 0.63 (0.36, 1.09)	0.224 0.099
	**Model 2**	1.82 (0.25, 13.31)	0.558	0.28 (0.06, 1.25)	0.095	***TL*** 2.88 (0.24, 34.21) ***PR*** 0.28 (0.07, 1.08)	0.402 0.064
**Lithuania**	**Model 1**	5.48 (0.41, 73.12)	0.198	0.34 (0.05, 2.44)	0.285	***TL*** 5.26 (0.37, 75.07) ***PR*** 0.34 (0.04, 2.63)	0.221 0.299
	**Model 2**	3.98 (0.28, 56.92)	0.309	0.33 (0.04, 2.91)	0.317	***TL*** 3.85 (0.25, 59.96) ***PR*** 0.33 (0.04, 3.09)	0.336 0.331
**Whole cohort**	**Model 1**	2.56 (0.72, 9.13)	0.148	0.63 (0.38, 1.05)	0.074	***TL*** 3.25 (0.82, 12.85) ***PR*** 0.61 (0.36, 1.01)	0.093 0.055
	**Model 2**	2.49 (0.52, 11.78)	0.252	0.36 (0.13, 1.03)	0.057	***TL*** 3.61 (0.60, 21.56) ***PR*** 0.35 (0.13, 0.97)	0.160 0.042

Results of the logistic regression model analysis with the presence of severe retinopathy (proliferative retinopathy/status post laser photocoagulation) as the response variable. Data are presented as odds ratios with 95% CI and *p*-values. Model 1 was adjusted for age, sex, and BMI; Model 2 was adjusted for age, sex, BMI, diabetes duration, waist/hip ratio, HbA1c, arterial hypertension, total cholesterol, high-density lipoprotein cholesterol, and triglycerides. Models in the whole cohort analysis were adjusted for participating countries as well. TL—relative telomere length; PR—serum proteasome concentration.

## Data Availability

The data underlying this article are available in the article and its online Appendix A.

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
