# Peer review of "Telomere Lengths and Serum Proteasome Concentrations in Patients with Type 1 Diabetes and Different Severities of Diabetic Retinopathy in Latvia and Lithuania"

_jcm, 2022, doi:10.3390/jcm11102768_

Round 1

Reviewer 1 Report

In this study, Zane Svikle et al. detected the telomere length and circulating proteasome concentration in patients with different stages of diabetic retinopathy in Latvia and Lithuania, and analyzed the differences in correlation patterns of anthropometric and clinical biomarkers with telomere length and proteasome concentration between two different stages of diabetic retinopathy.

While, the design and analysis of this study have the following problems:

1. The grouping of subjects included in the study was chaotic.

Patients with no diabetic retinopathy is T1D group? The inclusion criteria and exclusion criteria for recruited patients with T1D were not clear. According to which standard?

Since the purpose of the study is compare telomere length and circulating proteasome concentration in patients with T1D and different stages of DR, the analysis has no for T1D group.

2. Why included NDR and NPDR into one group, and included PDR and LPC into one group? NDR is T1D group, or diabetic group? Does laser-photocoagulation therapy affect circulating proteasome concentration?

3. Why use spearman correlation for all variables?

4. In NDR/NPDR group, the body mass index and waist/hip ratio were significantly negative correlated with telomere length. Is there any link between telomere length and obesity?

Author Response

Dear reviewer, thank you very much for your valuable comments which helped us to improve the manuscript. Please find the responses to reviewer comments below. We have addressed all questions raised, and hope that the manuscript is now acceptable for publication. For easier reviewing, reviewer comments and our answers were organized in the table (attached) 

We confirm that all authors of the manuscript have reviewed the revised manuscript and agreed to its contents. The author list has not been changed.

Reviewer 2 Report

The manuscript of Svikle et al. investigates telomeres length and circulating proteasome concentration in diabetic retinopathy and type 1 diabetes patients. The manuscript is nicely written, results are clearly presented and well discussed. My only major concern is whether the results should not be reanalysed according to the gender of the patients, that was described to influence telomere length, and reflect this in the results description and discussion.

Author Response

Dear reviewer, thank you very much for your valuable comments which helped us to improve the manuscript. Please find the responses to reviewer comments below. We have addressed all questions raised, and hope that the manuscript is now acceptable for publication. For easier reviewing, reviewer comments and our answers were organized in the table (attached). 

We confirm that all authors of the manuscript have reviewed the revised manuscript and agreed to its contents. The author list has not been changed.

Round 2

Reviewer 1 Report

Why use spearman correlation for all variables? 

Author explained in the “Statistical analysis”: “Most of the variables analysed did not correspond to the normality assumption (according to ShapiroWilk test). However, it means some variables correspond to the normality assumption? Why not use Pearson correlation for the analysis of these variables?

Author Response

Rebuttal letter to reviewer 1, round 2

Dear reviewer, thank you very much for your question. Our answer is below.

Question: Author explained in the “Statistical analysis”: “Most of the variables analyzed did not correspond to the normality assumption (according to ShapiroWilk test). However, it means some variables correspond to the normality assumption? Why not use Pearson correlation for the analysis of these variables?

Answer: All variables were re-examined for normality, taking into account both the subgroups of retinopathy and the countries. Although the assumption of normality could not be rejected for several clinical parameters, serum proteosome concentration and telomere length did not follow a normal distribution in all cases considered in the correlation analysis (Shapiro-Wilk test p-values<0.001). As correlations were calculated between the clinical parameters and/or serum proteosome concentration/telomere length, there were no pairs of variables where both of them met the normality assumption. For this reason, Spearman's correlation coefficient was used throughout the correlation analysis.